# Single-cell RNA-seq reveals disease-specific CD8+ T cell clonal expansion and a high frequency of transcriptionally distinct double-negative T cells in diabetic NOD mice

Md Zohorul Islam[1,2,3,4◦], Sam Zimmerman[1,2◦], Alexis Lindahl[5], Jon Weidanz[6,7], Jose Ordovas-Montanes[8,9], Aleksandar Kostic[1,2*], Jacob Luber[10‡*], Michael Robben[5,10‡*]

**1** Section on Pathophysiology and Molecular Pharmacology, Joslin Diabetes Center, Boston, Massachusetts, United States of America, **2** Department of Microbiology, Harvard Medical School, Boston, Massachusetts, United States of America, **3** Section of Experimental Animal Models, Department of Veterinary and Animal Sciences, University of Copenhagen, Copenhagen, Denmark, **4** CSIRO Health & Biosecurity, Australian Centre for Disease Preparedness, Geelong, Victoria, Australia, **5** Department of Animal Science, University of Illinois, Urbana-Champaign, Illinois, United States of America, **6** Department of Kinesiology, The University of Texas at Arlington, Texas, United States of America, **7** Department of Bioengineering, The University of Texas at Arlington, Texas, United States of America, **8** Division of Gastroenterology, Boston Children's Hospital, Boston, Massachusetts, United States of America, **9** Harvard Stem Cell Institute, Harvard University, Boston, Massachusetts, United States of America, **10** Department of Computer Science and Engineering, The University of Texas at Arlington, United States of America

◦ These authors contributed equally to this work.
‡ These authors also contributed equally to this work.
\* robben@illinois.edu (MWR), jacob.luber@uta.edu (JML), aleksandar.kostic@joslin.harvard.edu (ADK)

## Abstract

T cells primarily drive the autoimmune destruction of pancreatic beta cells in Type 1 diabetes (T1D). However, the profound yet uncharacterized diversity of the T cell populations in vivo has hindered obtaining a clear picture of the T cell changes that occur longitudinally during T1D onset. This study aimed to identify T cell clonal expansion and distinct transcriptomic signatures associated with T1D progression in Non-Obese Diabetic (NOD) mice. Here we profiled the transcriptome and T cell receptor (TCR) repertoire of T cells at single-cell resolution from longitudinally collected peripheral blood and pancreatic islets of NOD mice using single-cell RNA sequencing technology. We detected disease dependent development of infiltrating CD8+ T cells with altered cytotoxic and inflammatory effector states. In addition, we discovered a high frequency of transcriptionally distinct double negative (DN) T cells that fluctuate throughout T1D pathogenesis. This study identifies potential disease relevant TCR sequences and potential disease biomarkers that can be further characterized through future research.

## Introduction

The NOD mouse model has been demonstrated to be a translational model for T1D in humans, sharing many common markers of disease progression. Similar to humans, the onset

**Data availability statement:** All required data are incorporated in the paper, supplementary materials, and GitHub repository. Additional information or data requests should be directed to the corresponding author. The single-cell gene expression data were submitted to the Gene Expression Omnibus (GEO) database with an accession number of GSE200695 (https://www.ncbi.nlm.nih.gov/geo/query/acc.cgi?acc=GSE200695). Bioinformatic pipelines and code used in this study are available on GitHub at https://github.com/szimmerman92/NOD_t1d_islets_pbmc_single_cell_tcr.

**Funding:** This study was supported by grants from the Beatson Foundation (Grants ID 2800013). The salary of the author MZI was supported by Lundbeck Foundation, Copenhagen, Denmark (Grants ID R288-2018-1123). This includes all of the funding sources that supported this grant and there were no additional external funding sources.

**Competing interests:** The authors have declared that no competing interests exist.

of T1D in the NOD mouse is due to destruction of pancreatic β-cells in a multi-stage progression of autoimmune reactions. Islets remain free from lymphocytic infiltration in NOD mice until 3-4 weeks of age [1],[2]. At around 12 weeks, self-tolerance becomes broken due to the imbalance of regulatory T cells (Tregs) and effector CD8 + T cells [3]. Insulitis in NOD mice involves many types of lymphocytes and myeloid cells [4,5]. However, the T1D pathogenesis in this model is primarily driven by T cells [6–8]. It has been previously shown that CD4 + and CD8 + SP T cells are both needed for autoimmune destruction of β-cells [9–11].

β-cell reactive T cells that target autoantigens like GAD, IGRP, and insulin antigen are commonly seen in T1D development [12–16]. Other cell types such as B cells, dendritic cells, macrophages, and NK cells can also be found at the site of insulitis [17,18]. Once formed, insulitic lesions are continually replenished by new lymphocytes from peripheral circulation [19]. Therefore, T cell recruitment to the islets is considered a continuous process. Development of T1D in a mouse can be predicted from the detection of circulating autoreactive T cells in peripheral blood [15,20,21].

T cell receptor (TCR) repertoire profiling is a powerful tool for identifying T cell clones and phenotypes directly linked to insulitis and β-cell destruction. The majority of TCR profiling studies characterize circulatory T cell clones [22–25] with few studies looking at islets or other lymphoid organs, [26,27]. Using this method, autoreactive T cells in blood and islets have been observed in subsets of memory T cells [28,29] and T regs [30]. Marrero et al. [29] analyzed islet-infiltrating memory CD4 + T cells in prediabetic and recent onset diabetic NOD mice. They found many unique TCR clonotypes in islet-infiltrating CD4 + T cells and noted that TCR β repertoires were highly diverse at both stages of T1D development.

Recent technologies have enabled profiling the transcriptome at single-cell resolution, which has resulted in the growth of single-RNA sequencing studies of T1D tissues over the last few years [18,31–38]. Similar technologies enable the sequencing of TCR rearrangements at the level of individual clones, which can be collected concomitantly with RNA-sequencing data [39]. In this study, we use single cell TCR and RNA sequencing (scTCR-seq and scRNA-seq) to profile the transcriptional activity of autoreactive clones in pancreatic islets and PBMCs of diabetic NOD mice. We identify an uncharacteristically high proportion of DN T cells in both islets and blood of NOD mice. We also find that invading CD8 + T cells in diabetic NOD mice acquire a more exhaustive phenotype than the same cells in non-diabetic NOD mice. In this analysis, we identify new NOD TCR clones that are disease specific and not found in other studies that can be explored in future experiments.

## Results

### DN T cells are expanded in NOD mice and follow CD4/CD8 like lineages

We conducted paired single-cell RNA sequencing (scRNA-seq) and TCR repertoire sequencing on T cells collected from the peripheral blood and pancreatic islets of NOD mice (Fig 1a). We monitored female NOD mice from the age of 3 weeks to 40 weeks for the natural onset of diabetes (two consecutive blood glucose readings ≥ 250 mg/dl). Diabetic onset occurred in 70% of NOD mice by 40 weeks of age without any significant difference in body weight between diabetic and non-diabetic groups (Fig 1b-c, S1 Table in S1 File). There was no significant difference in T cell frequency in diabetic and non-diabetic mice blood (Fig 1d, S2 Table in S1 File).

T cells were enriched from PBMC and pancreatic islets and pooled together into six pools of 3 phenotypically similar mice before being sent for single cell library preparation (S1 Table in S1 File). In addition, four pooled PBMC samples from non-diabetic mice were collected at 34 weeks of age and included for comparison with PBMC of diabetic mice. T cell extractions

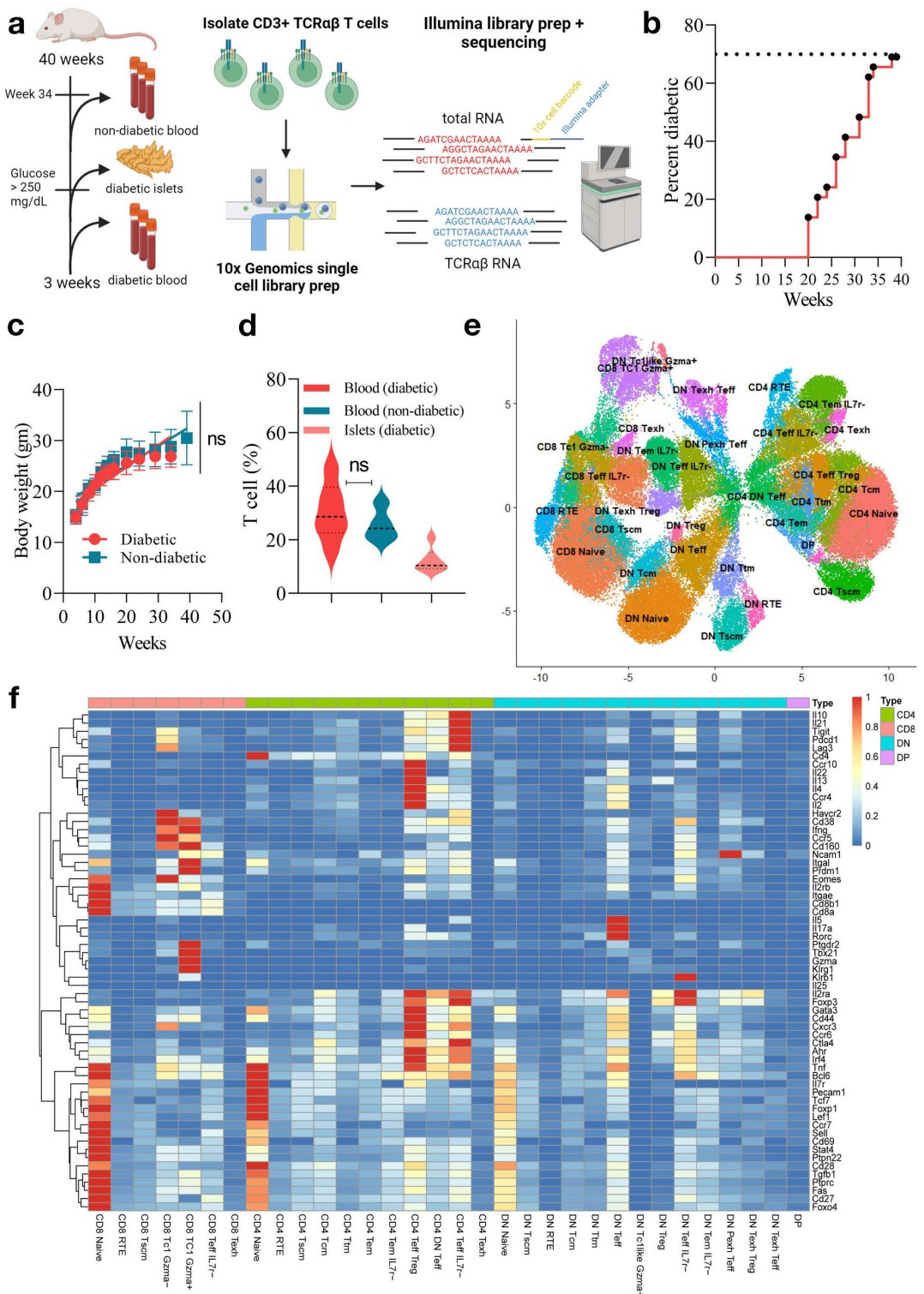

**Fig 1. Single-cell RNA-seq analysis of peripheral blood and islet-infiltrating T cells from NOD mice.** (a) Experiment design for single-cell analysis of T cells and TCR clonality detection from peripheral blood and islets of NOD mice, including magnetic separation of CD3 + T cells, 5'-scRNA-seq, and V(D)J profiling, and computational approaches for TCR matching clone detection. (b) Diabetes incidence up to the study endpoint of 40 weeks. Two consecutive blood glucose readings ≥ 250 mg/dl were considered for the onset of diabetes. (c) No significant changes in body weight on average were observed in

diabetic mice prior to sacrifice. (d) Frequency of T cells among all cells in peripheral blood and islets. T cell frequency in 40 week non-diabetic islets was < 1%. T cells were isolated by magnetic separation using monoclonal anti-mouse CD3ε antibodies conjugated to Biotin. (e) UMAP visualization of T cell populations in NOD mice. T cells from the peripheral blood of diabetic mice (n = 46,741 cells) at the onset of diabetes, peripheral blood of non-diabetic NOD mice (n = 28,457 cells) at 34 weeks of age, and islet-infiltrating T cells (n = 27,650 cells) of diabetic mice at the onset of diabetes were integrated using SCTransform. (f) Pseudo Bulk expression of canonical marker genes of T cells. Identified clusters represented by column are arranged first by their respective compartments (CD8+, CD4+, DN, DP) and then from left to right in order of T cell maturation.

from non-diabetic mouse pancreatic islets resulted in too few cells for scTCRseq. As a result, we obtained sequencing reads from a total of 102,848 high quality peripheral blood and islet derived T cells from both diabetic and non-diabetic mice (S2 Table in S1 File). Among filtered cells, 97% positively expressed CD3e using Seurat standards for positive expression.

T cells were clustered based on variable gene expression and immune marker weighted principal components that increased the biological variability observed between cells (S1 Fig a-d). In total, 32 clusters were identified (Fig 1e) and annotated based on expression of immune markers in each cell (S1 Fig e, S3 Table in S1 File) and pseudobulk expression patterns (Fig 1f, S2 Fig, S4 Table in S1 File). Expression patterns of the most common T cell markers revealed surprising results about the T cell compartment (Fig 2a-d). Primarily, an uncharacteristically high rate (~33%) of double negative (DN) T cells in all tissues (Fig 2e), which lacked expression of either CD4 or CD8. Abnormal proliferation of DN T cells in NOD mice are consistent with previous reports [40]. DN populations were confirmed not to express γδ receptor or NKT markers (S3 Fig a). There was also no correlation between CD3 expression, nCount or nFeatures and CD4/CD8 expression, suggesting that the identification of significant amounts of DN T cells was not due to sampling/sequencing error (S3 Fig b-i).

Clusters seemed to predominantly represent the effector, memory, and exhaustive states of invading and circulating T cells. Within effector populations, less than 20% of cells expressed markers associated with known subsets of CD4+ T cells, however, of that 20% there was greater enrichment of Th1-like CD4+ T cells in islet infiltrating populations (Fig 2f, S4 Fig a-c). We applied pseudotime trajectory analysis through Monocle3 which revealed a linear relationship from naive to effector cells with memory T cells developing throughout maturation (Fig 2g, S4 Fig d). According to the trajectory, it may be likely that there is transition between SP and DN cells at both the naive and effector states. Effector T cells correlated highly with pancreatic islets while phenotypically exhausted T cells, which were higher in the blood, were found to occur at later stages in pseudotime differentiation (S4 Fig e-f). Diabetic mice showed higher enrichment of exhausted (Texh) and pre-exhausted (Pexh) DN T cells in the blood, but equivalent amounts of exhausted CD4+ SP/CD8+ SP T cells and CD8+ Short Lived Effector Cell-like (SLEC-like, Il7r-/Gzmb+) cells (Fig 2h).

## DN T cell populations change significantly during diabetic progression in the NOD mouse

To validate elevated levels of DN T cells in the NOD mice we performed flow cytometry and meta-analysis of comparable datasets. First, we stained pooled splenocytes and thymocytes from 24 week non-diabetic and diabetic NOD mice with markers for CD3, CD4, CD8b, FasL, B220, TCRB, and NK1.1 (S5 Fig a). We were surprised to see that CD3+ DN splenocytes made up greater than 25% of CD3+ T cell populations in the spleen (S5 Fig b). Consistent with reports from MRL/lpr mice [41], > 30% of the DN T cells were B220+, however, they did not express more FasL or Fas than CD4+ or CD8+ T cells. As expected, none of these DN T cells

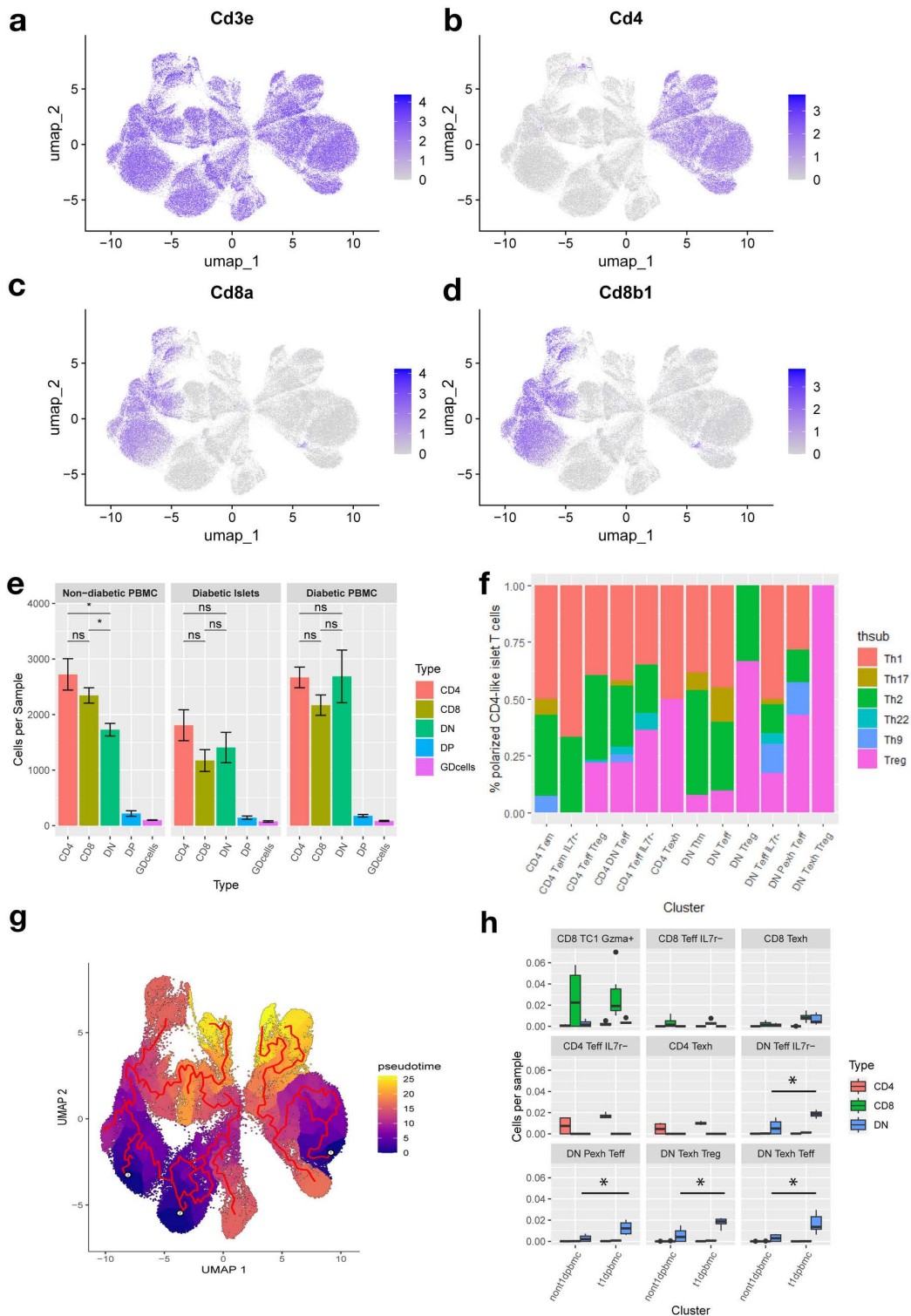

**Fig 2. Single cell transcriptomic data shows independent development of DN, CD4, and CD8 T cells in diabetic NOD mice.** (a-d) Feature plots visualizing the expression of major T cell markers CD3, CD4 and CD8 show a large proportion of DN T cells. (e) Bar graph representing the normalized total counts of predicted T cells in each tissue. T cell phenotype was determined for each cell by gating between major populations of CD4 or CD8 (CD8a and CD8b1) RNA expression. Lack of marker expression was not correlated with poor sampling of features or low CD3 expression. γδ T cells (GDcells)

were gated on expression of the only detectable γδ TCR gene (Trgv2). (f) CD4 + subset as a proportion of all detected cells in a cluster with a subset identity. Subsets were identified through gating on Infγ, IL-4, Foxp3, IL-17, IL-10, and TNFA. (g) Monocle3 trajectory shows linear progression from naive state T cells to effector cells and exhausted T cell populations. Pseudotime and trajectory correlated with predicted memory T cell origin (Tscm, Tcm, Tem, etc…). (h) Normalized proportion of CD3 + T cells belonging to Pexh and Texh clusters compared between diabetic and non-diabetic blood samples. Statistical significance of Wilcoxon ranked sum test reported. * $p < 0.05$.

bore the NK1.1 marker for NKT cells, however we did observe reduced MFI for TCRβ on the cell surface. We have confirmed through additional analysis that this did not correlate with an increase in gamma delta TCR abundance (S5 Fig c-d). Thymic populations had relatively fewer CD3 + DN T cells (<20%), however the populations were still largely B220$^+$ and TCRβ$_{low}$ but had greater expression of FasL and NK1.1 than CD4 + and CD8 + populations (S5 Fig e,f).

We were interested to see how populations of DN T cells changed over time in diabetic mice, so we conducted flow cytometry on paired blood samples from diabetic and non-diabetic mice used in single cell analysis at 4 different time points (Fig 3a). We have determined that the DN T cell compartment in the blood is negatively correlated with CD4 + populations but not correlated with CD8 + populations that increase steadily over a 24-week period (Fig 3b). Diabetic condition results in a shift from mostly CD4 + T cells to mostly DN T cells by week 24, whereas CD4 + T cells remain slightly more abundant throughout the same time frame in non-diabetic mice (Fig 3c). Normalizing diabetic mice by hyperglycemic onset (date at ≥ 250 mg/dL) shows an opposing sinusoidal trend between CD4 + and DN T cells in the blood resulting in a dip in CD4 + T cells and spike in DN T cells approximately 30 days before detectable increases in blood sugar (Fig 3d). This reversal was not present to the same extent in non-diabetic mice. We cannot conclude about the proportion of DN T cells in the pancreas over the NOD lifespan because they must be sacrificed at only one time point. However, the total number of circulating T cells remains level which means they must be coming from activated splenic populations or recirculated from inflammatory tissues.

In a similar NOD mouse scRNA-seq dataset [42], we identified similar proportions of DN T cells that were not previously identified in that study (Fig 3e-i). Of the T cells misidentified in the previous study, more DN T cells were misattributed as CD4 + T cells than CD8 + (S6 Fig a). Genes with distinct expression patterns in our dataset had similar patterns of expression and previously annotated resting Tregs were found to be analogous to the DN Treg cluster in our dataset (S6 Fig b, c).

## High enrichment of lipid metabolism genes in islet localized effector T cells

Differential expression analysis shows few differences between comparable populations of DN and CD4 + /CD8 + T cells (S5 and S6 Tables in S1 File). Effector cell transition for DN, CD4 +, and CD8 + T cells resulted in marked upregulation of many common genes including Carboxypeptidase a1 (*Cpa1*), *Cpb1*, Carboxyl Ester Lipase (*Cel*), and Chymotrypsin c (*Ctrc*) which are involved with metabolic pathways that are attributed to pancreatic inflammation (S7 and S8 Tables in S1 File). Progression to Texh phenotypes and removal from pancreatic islets for DN and CD4 + cells led to upregulation of markers associated with immunosuppression and exhaustion like *TIGIT* and *CTLA4* (S9 Table in S1 File) and CD8 + T cells gain expression of *Gzma*, *Gzmb*, Killer cell Lectin-like Receptor G1 (*Klrg1*), Sphingosine-1-phosphate receptor 5 (*S1pr5*), Zinc finger E-box-binding homeobox 2 (*Zeb2*), Galectin-3 (*Lgals3*), and Glucagon (*Gcg*), and lost expression of *Ctrb1*, CXC Chemokine Receptor 5 (*Cxcr5*), and *CD27* (S10 Table in S1 File).

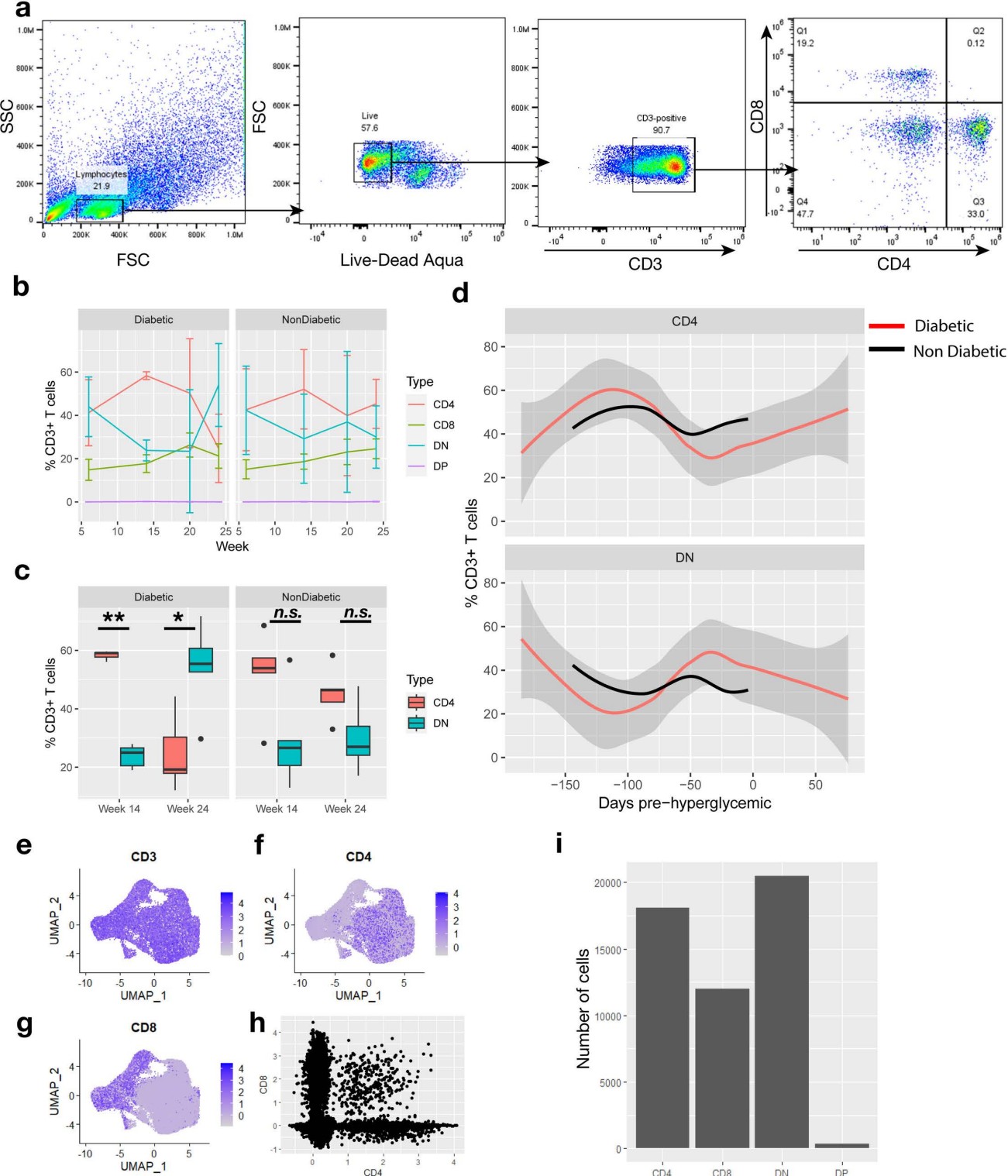

**Fig 3. DN T cell population is dynamic throughout disease progression and detectable through flow cytometry and external scRNA-seq data.** Flow cytometry (a) was performed on CD3 + enriched blood samples extracted from diabetic and non-diabetic mice at 4 time points using antibodies for CD3e, CD4, CD8a, and CD8b. (b) Percent of total live CD3 + population was calculated for SP, DN, and DP subsets in both diabetic and non-diabetic PBMCs. (c) A significant change in the proportion of CD4 + and DN cells is detected at weeks 14 and 24 in the diabetic mice but not the non-diabetic mice. Statistical significance of t-test is reported. *p < 0.05. **p < 0.01. (d) For each diabetic mouse, flow measurements were recalculated based on the

timing of detectable hyperglycemia and plotted as a Loess smoothed average over the 200 days preceding diabetic onset in red. Confidence intervals are represented using gray spline volume. Flow measurements were recalculated in non-diabetic mice using the average day of diabetic onset (151) and a Loess smoothed average was plotted in black. Data from (Collier et al., 2023) was downloaded and re-analyzed for CD4 and CD8 expression (e-h). (g) Cells with positive expression of CD4 but negative for CD8 were annotated as CD4 and vice versa for CD8, with DN cells being labeled for cells with negative expression of either CD4 or CD8 and vice versa for DP.

We observed expected expression patterns of common T cell genes like IL2ra, GATA3, and Foxp3 (S6 Fig a). Expression of the survival signal marker IL7R decreased in Tpex and Tex populations (Fig 4a) and to a greater degree in DN clusters (Fig 4b) which is consistent with previous results. Increased expression of Klrg1, Gzma, and Perforin (Prf1), and decreased expression of IL7r suggest that effector CD8 + T cells become Short Lived Effector Cells (SLEC) and accumulate in the blood due to an increase in expression of S1pr5 [43] (Fig 4c). Expression of immune checkpoint proteins like Programmed Death 1 (*PD1*), *TIGIT*, Cytotoxic T-Lymphocyte Associated protein 4 (*CTLA4*), and T cell Immunoglobulin and Mucin domain 3 (*TIM3*) were increased in exhausted populations (S7 Fig b).

We confirmed that loss of *IL7R* and increase of exhaustion markers are correlated with aromatic and organic cyclic pathway gene expression (S11 and S12 Table in S1 File). Clusterwise Gene Set Enrichment Analysis (GSEA) (Fig 4d) led to the discovery that DN specific resting Tregs and DN effector cells experience increased expression of Hypoxia Induced Factor 1a (HIF1a) and the Glucocorticoid receptor (NR3C1) (S7 Fig c), and CD4 + SP/DN effector cells are heavily enriched for lipid metabolism genes. A module of coexpressed genes specific to granzyme regulation (MEYellow) is found enriched and expressed at higher rates in islet infiltrating T cells (Fig 4e). The yellow module was highly correlated with zymogen activation genes, which is seen in suppression of CD8 + GzmA/B expression, and lipid metabolism genes (Fig 4f, S7 Fig d). The majority of lipid metabolism gene expression was due to Cel and Pancreatic Lipase Related Protein (PNLIP) expression which was specific to CD3 + T cells (S7 Fig e).

## Clonal specificity of KLRG1 + IL7R- CD8 + T cell development in diabetic mice results in Gzma + cytotoxic T cells trafficked out of the pancreatic islets.

V(D)J profiling resulted in 85,321 unique α and β TCR sequences across 92,180 cells (S13 Table in S1 File). There were 82,008 TCR sequences found in only a single cell, 990 sequences found in 3 or more cells, and 79 sequences found in more than 10 cells (S8 Fig a-b). The largest number of cells found with a single TCR was 75. Of TCRs with greater than 3 cells, 457 were found in at least 2 different samples and 130 were found across diabetic and non-diabetic mice (S8 Fig c-d). Highly expanded clones (>10 cells) clustered into 3 groups, CD8 + Gzma-, CD8 + Gzma +, and effector CD4 + T cells (S8 Fig e). We found that neither diabetic tissue had many clones that were in the Gzma + group which were also found to be Klrg1 + upon differential expression analysis (S8 Fig f and S14 Table in S1 File). CD8 + cells that are Gzma + and Klrg1 + are often considered to be Short Lived Effector Cells (SLEC) so we will consider these to be SLEC-like. The diabetic tissues were more enriched in non-SLEC like Il7r- Gzma- CD8 + T cells and DN/CD4 + SP T effector cells that were upregulated in exhaustion markers like Ctla4, Lag3, and Tigit (S8 Fig g and S4 Table in S1 File).

Analyzing paired blood and islet samples from diabetic NOD mice, we were able to trace genotype specific clones of islet-infiltrating T cells in the blood (Fig 5a). We found a total of 462 TCR clones belonging to infiltrating T cells (S15 Table in S1 File). This resulted in 2,118 total T cells infiltrating TCR sequences, with 51 of the cells from Nondiabetic PBMCs and

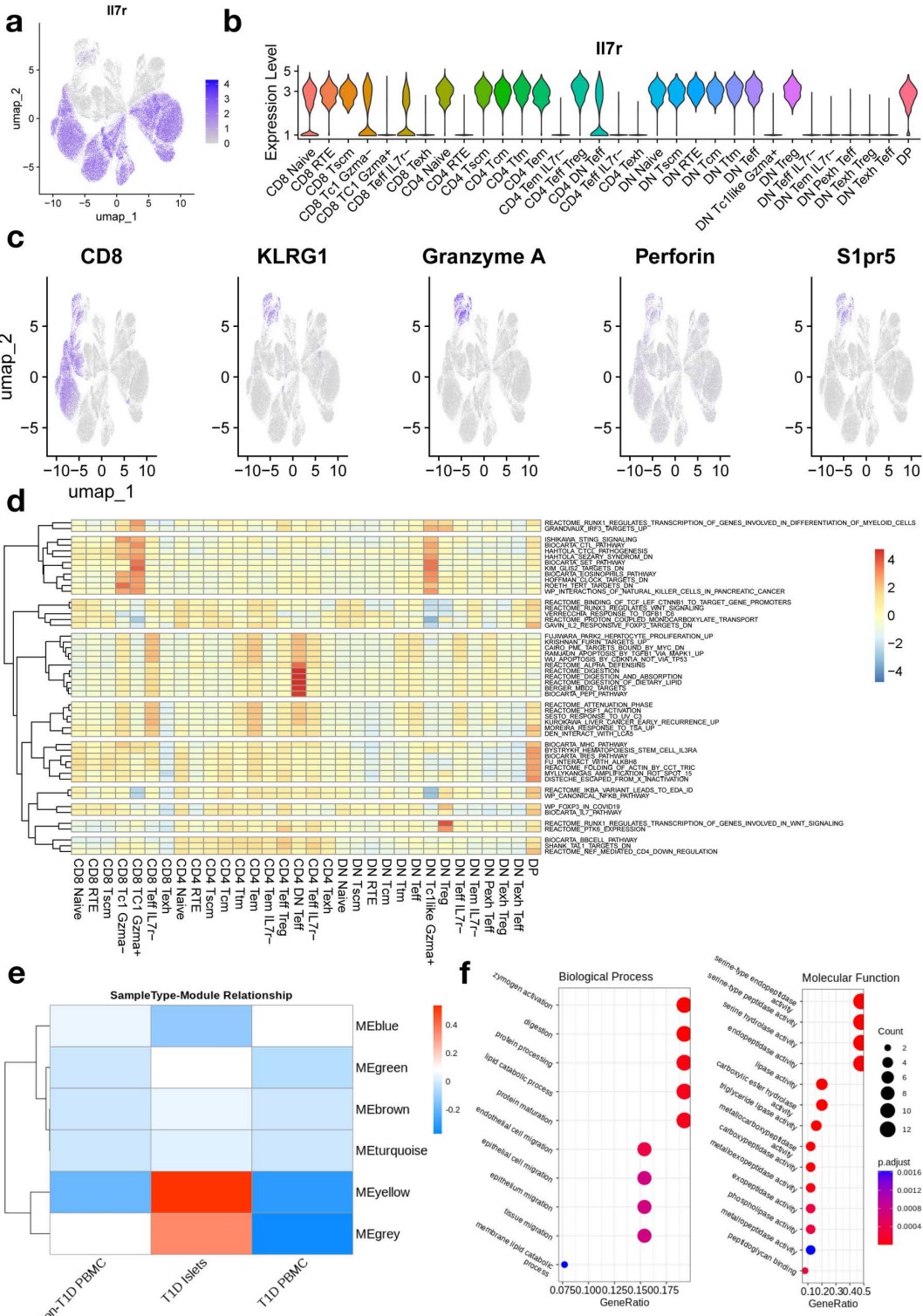

**Fig 4. Gene and pathway expression among T cell clusters involved in diabetic response.** (a-b) Expression IL7r T cell survival gene across T cell clusters. (c) Non-exhausted T cell cluster shows SLEC-like expression patterns, shown through Klrg1 + Gzma + IL7r- status. Expression of S15 is specific to SLEC-like clusters. Scale for color values is shown in Figure 1a. (d) Aggregate enrichment of mouse curated gene sets (MSigDB, downloaded 3/08/2024) in each T cell cluster. (e) Tissue specific enrichment of WGCNA modules. GO enrichment of genes predicted in the yellow module.

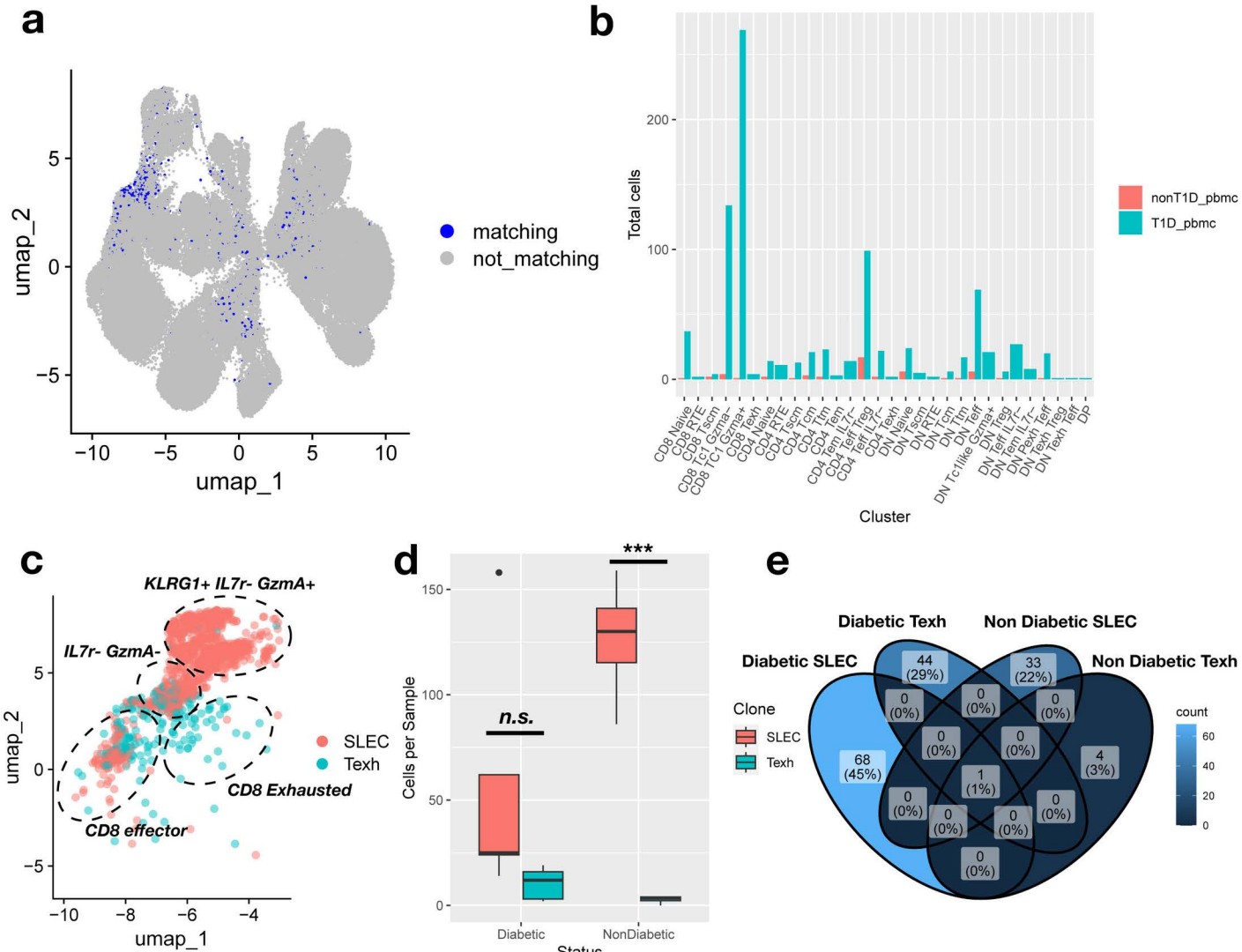

**Fig 5. Clonal expansion of invasive T cells in NOD mice reveal diabetic specific CD8 + TCR and interesting DN development pathways.** (a) Matching (infiltrating) status for T cells receptor sequences was predicted based on alpha and beta CDR3 sequences in each cell. (b) Diabetic mice contain more infiltrating clones in the circulating T cell population than non-diabetic. (c) Lineage tracing shows that CD8 + TCR clones exclusively transition from an effector state to SLEC-like (147 TCR sequences) or Texh (58 TCR sequences). Cells from clonal lineages are plotted in UMAP space and colored for their clonal identity. Labels are added to specify the cluster identity of cell populations. Only one clone was detected with cells in both populations. UMAP was cut off to focus on CD8 + populations, however clones were detected in DN and CD4 + clusters expressing either CD4 or neither marker. (d) The per sample number of T cells with SLEC-only or Texh-only clones were plotted by diabetic status of the pooled PBMC sample. Statistical significance is reported as results of a t-test between SLEC-only or Texh-only clones. ***p < 0.005. (e) We demonstrated that there were almost no common SLEC-like or Texh specific clones between diabetic and non-diabetic mice.

880 of the cells from Diabetic PMBCs. The greatest disparity between the diabetic and non-diabetic mice was the difference in the number of infiltrating Gzma positive and negative CD8 + T cells (Fig 5b).

Using lineage tracing, we identified TCR specific clonal lineages that progressed to a SLEC-like phenotype instead of an exhausted phenotype which was confirmed by lineage specific monocle3 trajectories (Fig 5c and S9 Fig a). Diabetic mice had nearly equal populations of SLEC-like and Exhausted CD8 + T cells while non-diabetic mice were over enriched for SLEC-like (Fig 5d). We found that exhausted CD8 + T cell clones were diagnosis specific

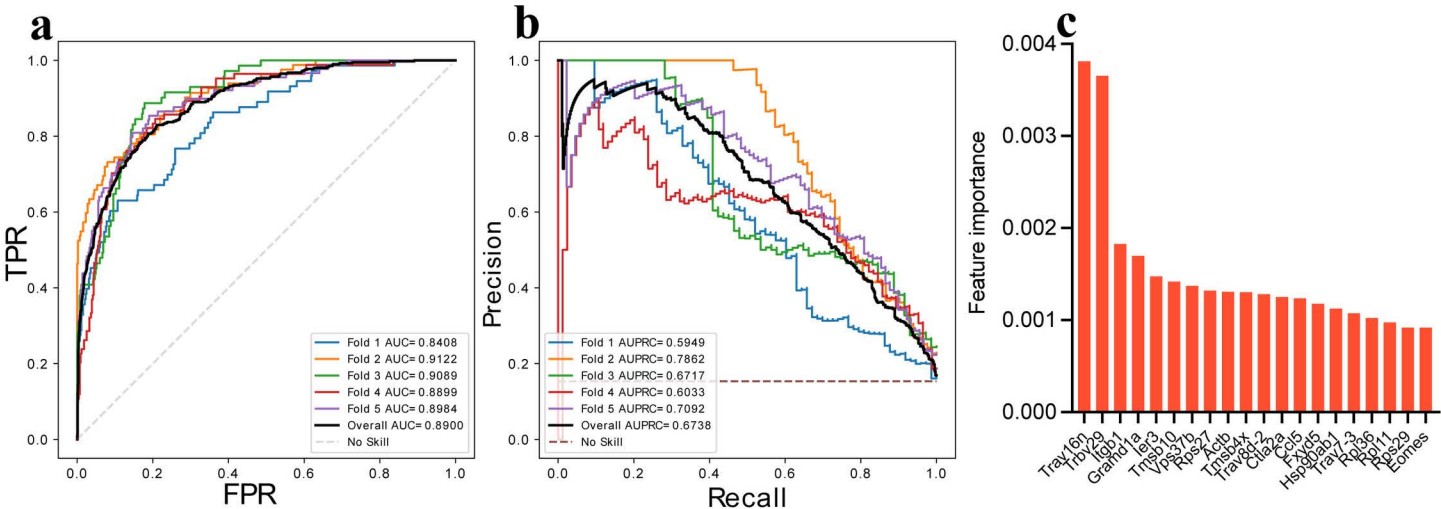

**Fig 6. The transcriptional signatures of T cells can predict the matching status of the T cell clone.** (a-b) Receiver operating characteristic curve and recall-precision plot with five-fold cross-validation showing the classification of cells as matching (infiltrating) or non-matching in the CD8 effector memory cells in blood based on logistic regression classifier. Each line represents training with a different batch of genes demonstrating that some genes were better predictors of infiltration. The dashed line is the metric accuracy for a random prediction with a 50% chance of success. Overall model performance is reported as Area Under the Curve (AUC). (c) A bar chart with the top 20 genes was found to be the most significant predictor of the matching status of the cells. The Y-axis shows relative importance for each gene on model prediction.

and represent a population of 44 unique TCRs that are only exhausted in the disease condition (Fig 5e).

Within the similar Collier et al., [42] dataset, we found that infiltrating T cells were more likely to be exhausted than non-infiltrating, and diabetic infiltrating had higher amounts of exhausted CD8 + and DN T cells (S6 Fig c). We found this consistent with our results which also showed an increase in exhaustion markers like Lag3 and TIGIT in infiltrating clones of diabetic mice (S9 Fig b). Differential expression analysis of infiltrating T cells in diabetic mice revealed them to be upregulated for inflammatory genes like Ifng (S9 Fig c, S16–S18 Tables in S1 File).

Because there is a question of the origin for the DN T cells, we sought to use clonally expanded cells to determine if there was transition occurring within single clones. Consistent with earlier evidence, we found that within clones, transition between T cell compartments was most likely to occur during DN and CD4 effector maturation with some transition occurring between naive cells (S9 Fig d).

To investigate whether identified T cell clonal populations were targeting previously known epitopes associated with diabetic progression we searched for exact CDR3 matches to the VDJdb, a database of experimentally validated TCR-pMHC interactions. We only found two hits of the expanded (>10) clones but among all clones we found 4,166 matching CDR3 sequences in the VDJdb (S19 Table in S1 File). Of the known epitopes sources to the matching CDR3 sequences, the most common was to Influenza A PA (686 hits) and PB1 (392 hits) with the second highest target being Murine Cytomegalovirus (MCMV) M45 (724 hits) along with recognition of epitopes from Lymphocytic Choriomeningitis Virus (LCMV), *Plasmodium Berghei*, and Respiratory Syncytial Virus (RSV). There were much fewer self-targeted antigens with the majority specific to Myelin Basic Protein (MBP) and Protein Arginine Deaminase 4 (PADI4), we did not observe any specific to known insulin specific epitopes. While the majority of the antigen specific T cells were evenly distributed among tissues and clusters, we

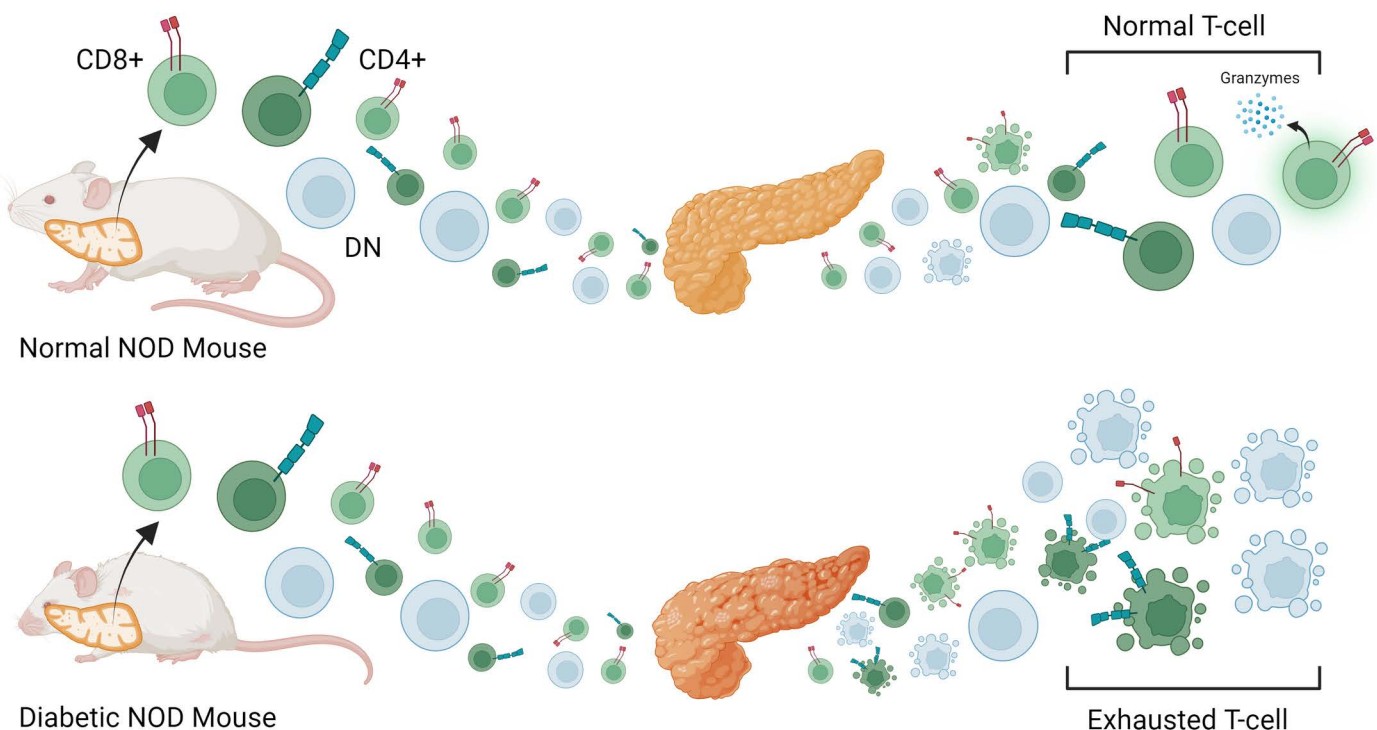

**Fig 7. Mechanism of T-cell immune regulation in NOD mice.** Autoreactive T cells that are either CD4+, CD8+, or DN are selected in the thymus and enter the bloodstream where they begin to mature into effector T cells that invade the pancreas. Under normal conditions T cells do very little damage to the pancreatic islets and return to circulation as normal effector and memory cells but in the diabetic mouse, damage is increased and leads to more T-cell exhaustion and reduced expression of GzmA in pancreas infiltrating CD8+ T-cells.

observed Influenza NP (ASNENMETM) targeting T cells had higher relative numbers in the pancreatic islets (S9 Fig e-f).

## V(D)J and immunoregulatory gene expression in T cell clones is associated with islet infiltration

Because T cell infiltration appeared to occur in specific populations, we predicted the infiltration status of T cell clones based on gene expression. We found that a regularized logistic regression classifier achieved moderately high sensitivity and specificity (Fig 6a-b, Overall area under the curve [AUC] = 0.89 and overall AUPRC = 0.67). Finally, we identified a list of genes that were found to be associated with the matching status of the cells (Fig 6c). Of the top 15 genes used in the matching status prediction, 4 are associated with clonal specificity in V(D)J rearrangement (*Trav16n*, *Trbv29*, *Trav8d*-2, *Trav7*-3), 6 are associated with immune cell function, and 5 are associated with ribosome function.

We looked at specific expression of TCR genes among clusters to see if specific TCR gene expression related to invasion and diabetes (S10 Fig). We do find an interestingly high amount of gamma delta TCR expression in the exhausted/overactive CD8+ and DN TC1-like cells, although they remain < 5% of all T cells. Reduced overall expression of TCR genes in the DN population is consistent with the $TCR\beta_{low}$ phenotype found through flow cytometry. Interestingly, the DN population has higher specificity for Trav15d-1 and both have higher expression of Trav16n which was identified to predict for islet infiltration. Trbv29 also seems to be more

specific for CD8 + T cells. Effector DN T cells have relatively higher expression of Trav15n and Trav15d than their CD4 equivalent (S10 Fig d). Based on entries in the VDJdb, Trav16n, which is focused to the IL7R⁻ T effector cluster is specific to the Influenza A PA (SSLEN-FRAYV) and PB1 (LSLRNPILV) antigens while Trav15d expressed in IL7R⁺ DN T effector cells has specificity to Plasmodium Berghei GAP50 (SQLLNAKYL).

## Discussion

T cell dysfunction has largely been shown to be the primary cause of diabetic progression in T1D patients and NOD mice [44]. In this model both CD8 + and CD4 + T cells show autoimmune potential and self-antigen recognition that leads to beta cell destruction in pancreatic islets. To explore how TCR restriction leads to T cell dysfunction we conducted paired scRNA and scTCR sequencing of T cell populations in adult female NOD mice. Because we cannot detect beta cell destruction in vivo, we sampled mice as soon as beta cell destruction was detectable through hyperglycemia but may have been censored from changes that occur early in disease progression.

A challenge we and many T cell researchers faced in single cell transcriptomics was a lack of meaningful biological variation between T cells for cluster assignment [45,46], which was evident due to high mixing of CD8 + and CD4 + T cells within clusters (S1 Fig b,d). We replicated a technique we have used previously to increase biological diversity in single cell datasets [47], by weighting biologically relevant markers more heavily than variably expressed markers in principal components. This led to biologically relevant clusters that fit current expectations of T cell maturation; however, it did likely produce some artifacts detectable in the UMAP which was a complaint in the original study. T cell subsets representing Th1, Th2, Th17, etc… were not apparent in clustering and further identification in Teff clusters based on common markers (IFNg, IL-10, IL-4, etc…) revealed on average that less than 20% of effector cells fit these classifications based on expression (S4 Fig b). These cell types may be hidden by the limitations of scRNA-seq where either low sampling produces zero inflated counts of marker genes or expression values do not reflect the production of subtype specific cytokines.

The most surprising outcome was identification of an over enriched DN T cell subset that was calculated to be ~ 30% of all CD3 + T cells. We have confirmed experimentally that these are CD3 expressing T-cells and are not NKT or γδ subsets. The DN T cell populations change over time coinciding with disease development. In wild type mice and humans, DN T cells are estimated at only 1-5% of CD3 + T cells [48]. However, increased DN populations have been reported in both NOD and MRL-*lpr* mice, a mouse model of inflammatory Systemic Lupus Erythematosus [40,49].

Studies in the lupus prone MRL-*lpr* mice, which contain a spontaneous mutation in the Fas receptor, have connected lymphoproliferation of DN T cells to a reduced ability to remove defective thymic or peripheral T cells [50]. The ability for the lpr and gld mutations to protect NOD mice from T1D correlated with our finding that diabetic mice undergo greater expansion of DN T cells, perhaps as protective mechanism of peripheral tolerance. The genetic similarities between these mouse models are limited as Fas has not been identified in any of the IDD loci and diabetic onset in NOD mice is likely polygenic [51].

We have confirmed similarities between *lpr* DN and NOD DN T cells, including increased amounts of B220 expression, even though there is seemingly no change in expression of the CD45 gene. These similarities may hint at a common development pathway, including likely peripheral development according to the single cell trajectory and clonal lineage tracing. Double negative T regs have also been shown to protect the NOD mouse from diabetes in adoptive transfer studies more than CD4 + Tregs [49]. Another potential reason that DN T

cells in the blood spike more intensely prior to hyperglycemia is that they are removed locally from the pancreas more than CD4 + T cells to block these protective effects. While it is unclear the exact effect that DN T cells may have on disease progression, we do observe increased DN exhaustion in the diabetic mice.

Through paired TCR sequencing, we observed the impact of CD8 + IL7R- KLRG1 + SLEC-like cytotoxic T cells in pancreas infiltrating lymphoid cells. We found that CD8 + T cell progression to SLEC-like or Texh phenotypes was clonally specific and leaned more towards Texh in the diabetic mouse. This might suggest that SLEC development is necessary for T1D protection which is consistent with reports that GzmA deficient NOD mice have enhanced disease development [52]. The diabetic exhausted clones identified could also be autoreactive TCRs that are more actively suppressed in the diabetic condition. Similarly, we found that infiltration into the pancreatic islets could be predicted using certain components of V(D) J recombination. It has already been suggested that the components we identified (Trav16n, Trbv29, Trav8d-2) are linked to autoreactivity of insulin peptides in NOD mice [53]. A majority of the genes specific to pancreatic islet localized T cells were related to lipid metabolism such as CEL and PNLIP, which have already been identified as mediators of pancreatic inflammation [54]. We did not identify any insulin specific peptides as targets of our dataset TCR, possibly due to a combination of few NOD experiments in VDJdb and poor CDR3 recognition in the TCR sequencing. We did find some targets which could have further impact on diabetic research, most notably a higher relative pancreatic infiltration of Influenza A NP targeted T cells, which previous studies have linked viral infection to greater incidence of T1D [55]. BLAST search of the epitope to self-antigenic proteins in mice resulted in a 100% match (NENMET) to Keratin 222 (KRT222).

## Conclusion

We characterized and identified a single-cell level T cell map in the peripheral blood and islets of NOD mice during the progression of Type 1 diabetes. Diabetic mice were found to have shockingly high levels of circulating and invading DN T cells and increased exhaustion of potentially immunosuppressive T cell subsets (Fig 7). DN T cell subsets increased during diabetogenesis suggesting that they proliferate more, die less, or are trafficked out of the pancreas more in concert with islet destruction. The primary difference in clonal expansion of diabetic mice was in CD8 + T cells. Diabetic mice showed greater expansion of clones that become exhausted as opposed to functionally active and SLEC-like. This is likely due to regulation by other immune cells in the pancreas (Tregs) or prior to infiltration of pancreatic islets. We identify previously uncharacterized clones associated with the regulation of CD8 + T cell exhaustion in the pancreas. It seems that in the NOD mouse, peripheral DN T cell proportion is a biomarker of diabetogenesis that can be used to detect diabetic onset as early as 14 weeks but whether a similar phenomenon is observed in humans must still be investigated.

## Materials and methods

### Experimental mouse model

We purchased three-week-old female NOD/ShiLtJ mice from The Jackson Laboratory (stock number 001976) and were monitored for natural diabetes onset until 40 weeks of age. The mice experiment was carried out in a specific-pathogen-free (SPF) facility at Joslin Diabetes Center (JDC) under standard housing, feeding, and husbandry. The mice experiment protocol was reviewed and approved by JDC's Institutional Animal Care and Use Committee (IACUC) (Protocol #2016-05). In addition, ARRIVE guidelines were strictly followed while conducting

the mice experiment. Briefly, approximately 200 µL of blood was collected through the lateral tail every two weeks. At the time of organ collection, mice were euthanized by isoflurane inhalation before cervical dislocation.

## Blood glucose measurement and diabetes diagnosis

Tail blood was used to measure glucose concentration. Every two weeks, blood glucose was measured by Infinity blood glucose test strips (GTIN/DI#885502-002000) and an Infinity meter. Mice showing two consecutive glucose readings of ≥250 mg/dl were considered diabetic [56,57].

## Blood lymphocyte collection and cryopreservation

Peripheral blood was collected from mice every two weeks. Approximately 200 ul of blood from each mouse was collected from the tail using a heparin coated Microvette tube to prevent blood clots (Sarstedt catalog#16.443.100). PBMC's were extracted using an equal volume of Histopaque 1083 (Sigma-Aldrich, Missouri, USA catalog#1083-1). Total cells count and viability were assessed on a Countess automated cell counter (Thermo Scientific,). After counting, the cell pellet was resuspended in 1 mL of Cryostor CS10 (Stemcell Technologies Catalog #07930) and frozen down for storage in liquid nitrogen.

## Single-cell collection from pancreatic islets and cryopreservation

Mice sacrificed at the onset of diabetes prior to detection of symptoms to prevent suffering and were perfused with collagenase (Vitacyte, CIzyme catalog #005-1030) and dissected to remove the pancreas for further processing. The pancreatic tissue was hand-shaken vigorously for 5-10 seconds and centrifuged at 500xg for 1.5 minutes then filtered to remove the remaining undigested tissue, fat, and lymph nodes. The filtrate was centrifuged at 500xg for 1.5 min at RT, and the tissue pellet was resuspended in 10 mL lymphocyte separation media (LSM) (Corning, cat #25072-CV) before purifying by density gradient centrifugation. Non-islet tissues were removed from the islet suspension by microscopic examination. These purified islets were subjected to non-enzymatic single-cell dissociation. The cell suspension was filtered through a 70µm cell strainer and counted before cryopreservation.

## T cell preparation for single-cell RNA-sequencing

PBMC and islets single-cell suspension preserved in liquid nitrogen were revived before the magnetic separation of CD3 + T cells. CD3 + T cells were isolated using the Miltenyi MACS CD3 microbead kit (Miltenyi catalog#130-094-973) following the manufacturer and 10x genomics recommended guidelines (10x Protocol CG000123 Rev B). The cell viability and count were performed before and after CD3 + T cell enrichment. The magnetically separated CD3 + T cells were placed on ice and immediately used for single-cell cDNA library preparation.

## Single-cell RNA and immune repertoire sequencing library preparation

Single-cell 5' gene expression (GEX) and V(D)J sequencing libraries of T cells were generated using the Chromium Next GEM Single Cell 5' Dual Index Reagent Kits v2 (10x Genomics, Pleasanton, CA, USA) following the manufacturer guidelines (10X protocol CG000331 Rev A).

## Sequencing of 5'GEX and V(D)J libraries

Gene expression and V(D)J libraries were sequenced on Illumina NovaSeq6000 sequencer by GENEWIZ (GENEWIZ, LLC, NJ, USA). Each of the 5'GEX and V(D)J libraries were prepared

with unique dual indexes, which allows multiplexing of all libraries for sequencing. Libraries were sequenced according to the 10X Genomics configuration. A minimum of 20,000 read pairs per cell were sequenced for 5'GEX libraries, and a minimum of 5,000 read pairs were sequenced for V(D)J libraries.

## Sequence demultiplexing and processing

Raw sequencing reads were processed using Cell Ranger version 6.0.2 to produce gene expression count matrices and TCR clonotype summary. Using the raw sequencing FASTQ files, we first run the *cellranger multi* pipelines for simultaneous processing and analysis of V(D)J and gene expression data. This pipeline generates single cell feature counts, V(D)J sequences, and annotations for a single library. The mouse reference genome mm10-2020-A was used for aligning gene expression sequences, and the mouse reference GRCm38 v5.0.0 was used for aligning V(D)J sequences.

## Analysis of single-cell gene expression data

All gene expression analyses were performed using R version 4.1.0 [58] and Seurat version 4.1.0 [59]. Other R packages used during this analysis include dplyr version 1.0.8, patchwork version 1.1.1, data.table version 1.14.2, ggplot2 version 3.3.5, cowplot version 1.1.1, viridis version 0.6.2, gridExtra version 2.3, RColorBrewer version 1.1.2, and tibble version 3.1.6. Seurat object was created individually for all samples using the barcode.tsv, features.tsv and matrix.mtx files generated from the *cellranger multi* pipeline. The min.cells parameter was set to 3 and the min.features parameter was set to 200 during the creation of Seurat objects. Cells were filtered out from the Seurat object based on several quality control parameters. First, low-quality cells were removed based on *CD3* gene expression (>0 counts), overall mitochondrial gene expression [60], and an aberrant high count of genes (200-5,000 features). Second, we filtered cells based on the expression of housekeeping genes (>0 counts).

The quality-filtered gene expression data were normalized and scaled by Seurat function *NormalizeData* and *ScaleData* with default "LogNormalize" parameters, generating log-transformed gene expression measurement per 10,000 reads. Weighted PCAs were calculated with weights applied to relevant immune markers taken from multiple sources [61–63] and 400 most variable features. SCTransform integration was applied to normalize libraries and PC 1-10 were used for KNN based Umap and Louvain clustering generation. Clusters were selected from the 1.5 resolution and hand annotated for cell identity. Monocle3 [64] was used to calculate pseudotime across clusters. Gating to identify CD4, CD8, and DN populations as well as Th subsets was done on normalized expression data using a linear gate between the median populations of cells (Supplementary Figure 4a) for each marker or a fixed value of 0.5.

## T cell clonal analysis

Matching clones of T cells between blood and islets were determined based on the similarity of TCR sequences. We only included the T cells with at least one alpha and one beta chain in the clonal analysis. The cells were assigned to a particular clonotype if they shared the same amino acid sequence of TCR alpha and TCR beta sequence. This same definition was also applied to cells that carry multiple alpha and beta sequences. Cells having identical TCR alpha or beta sequences were defined as matching clones between blood and islets. For example, if a T cell in blood carries identical TCR alpha or TCR beta sequence to a T cell in islets, it is defined as islet-matching T cell in blood or blood-matching T cell in islets. The total count of cells in a particular clonotype was used to determine the clonal expansion. TCR clonotype data were added to the GEX Seurat object as metadata for integrated gene expression and TCR

analysis. Predicted CDR3 sequences from T cells were aligned to the VDJdb database [65] to query for exact matches.

## Weighted co-expression analysis

Genes were clustered by similar patterns of expression across cells using the WGCNA package [66]. Co-expression clusters were generated using a soft power of 10 and a minimum module size of 10, with a merge cut height of 0.15. Clusters were randomly assigned colors to label modules.

## Gene set enrichment analysis (GSEA)

For GSEA, we downloaded the list of Hallmark pathways (v7.5.1) from the Molecular Signature Database (MSigDB, http://software.broadinstitute.org/gsea/msigdb/index.jsp) and used the ranked list of differentially expressed genes between different cell types. R package fgsea version 1.20.0 was used for GSEA.

## Machine learning

We first extracted the metadata, raw counts, and log normalized raw counts of gene expression from the integrated dataset to predict the matching status of cells based on gene expression profile. Then, we used logistic regression to make predictions with a liblinear solver and lasso penalty of 11. We used the Scikit-learn package in Python version 3.10.4 [67] for the regression analysis and used matplotlib to generate the plots.

**Flow cytometry.** For flow cytometry analysis, T-cell enriched samples from the blood of 5 diabetic and 5 non-diabetic NOD mice at 6,14,20 and 24 weeks of age were stained using labeled primary antibodies against CD4 (GK1.5, Thermofisher #12-0041-82), CD3e (17A2, Thermofisher #48-0032-82), CD8a (53-6.7, Thermofisher #56-0081-82), and CD8b (H35-17.2, Thermofisher #11-0083-82), NK1.1 (PK136, Thermofisher #407-5941-82), TCRβ (H57-597, BioLegend #109225), B220 (RA3-6B2, BioLegend #103212), FasL (MFL3, Thermofisher #63-5911-82), Fas (15A7, Thermofisher #46-0951-82), CD19 (1D3, Thermofisher #M004T02Y03-A), and TCRγδ (GL-3, Thermofisher #25-5711-82). Before staining, all samples were incubated with Live Dead Aqua Viability (Thermofisher #L34957) stain for 10 minutes in dark at 4°C. Samples were run on Attune NxT four color laser flow cytometer compensated with unstained control and single stained UltraComp eBead compensation beads (Invitrogen, #01-2222-41).

## Statistical analysis

All statistical analyses were performed with R or GraphPad Prism software, and P values < 0.05 were considered statistically significant. Figure legends show respective statistical analyses.

## Supporting information

**S1 Fig. Clustering and marker discovery.** (a) Clustering and UMAP of high quality T cells with principal components generated from top 2000 most variably expressed genes. (b) Expression of CD3, CD4, and CD8 across cells plotted on UMAP from variably expressed gene principal components. (c) Expression of CD4 and CD8a in each cell. Gating on normalized expression was performed at a positive value over 0.05 for CD4 or CD8a. (d) Gated cells represented as a proportion of CD3 + T cells in either clusters generated from variably expressed genes alone or variably expressed genes with T cell marker weighted principal

components. (e) Dotmap expression of T cell markers where the size of each dot represents the number of positively expressing cells and color is a continuous variable of average normalized expression across all cells.
(TIF)

**S2 Fig. Pseudobulk expression of genes used to identify and annotate clusters.**
(TIF)

**S3 Fig. DN T cell marker expression.** (a) Expression of Klrb1c (NKT cell marker) and Trgv2 (gamma delta T cell marker) plotted against the UMAP coordinate of each cell. Correlation of CD3e expression to (a) CD4 and (b) CD8a with red line representing a linear model of values. (d) Heatmap and (e) violin plot showing expression of CD3e across clusters demonstrates equivalent expression of CD3e. Expression of CD3e, CD4, and CD8a plotted against (f) nCount and (g) nFeatures for each cell. Red line represents a linear model for x and y values. (h) Correlation matrix plotted as a heatmap for expression of each marker with the linear correlation of (i) Cd8a and Cd8b demonstrating the heavy linkage between the two gene's expression.
(TIF)

**S4 Fig. Development of Effector T cell populations.** (a) Gating strategy example for identification of CD4 + T cell like subsets. T cell subsets were predicted as a lack of CD8 expression and concurrent expression of the following markers: Th1(Infg and TnfA), Th2 (IL-4 without IL-10 expression), Th17 (IL-17), Th22 (IL-22 and TNFA), Tfh (IL-4 and IL-10), Th9 (IL-10 without Ifng or TNFA expression), and Treg (Foxp3 and IL-10). T cells without those combination of marker expression were labeled NA. (b) Proportion of T cell subsets across clusters (average of 20% not NA found in each cluster). (c) Proportion of CD4 + T cell across CD4 and DN clusters separated by tissue. (d) Monocle clustering and trajectory in Seurat UMAP space. (e) Localization of islet T cells in UMAP space. (f) Average cells per sample per cluster faceted by disease condition and tissue source.
(TIF)

**S5 Fig. Validation of DN populations were performed using flow cytometry.** (a) Splenocytes were pooled from 24-week-old diabetic and non-diabetic NOD mice and then stained for T cell specific markers. (b) Boxplots show populations of CD3 + splenocytes of DN, CD4 + and CD8 + T cells. (c) Histogram of TCRB MFI shows lower protein abundance of TCR on the surface of DN T cells than CD4 + T cells. (d) Splenocytes largely are not expressers of gamma delta TCR. (e) Overall and (f) marker populations of CD3 + T cells in thymocytes.
(TIF)

**S6 Fig. DN T cells bioinformatic meta-analysis.** (a) Misclassification of DN T cells gated as described within this studies dataset. (b) Expression of markers IL7R and GzmA plotted on Collier et al., UMAP coordinates. (c) Average number of cells per sample per pre-annotated cluster faceted by disease and infiltration status and colored by the correct T cell compartment.
(TIF)

**S7 Fig. Marker and differentially expressed genes.** (a) Expression of commonly used markers in flow cytometric analysis of T cell populations. (b) Expression of exhaustion markers in T cells plotted in UMAP space. (c) Violin plot showing expression of Nr3c1 and Hif1a across clusters. (d) GSEA of grey module (MEGrey) using GO terms for biological process, cellular compartment, and molecular function. (e) Expression of Cel and PNLIPR across cells plotted in UMAP space.

(TIF)

**S8 Fig. Analysis of Infiltrating T cells in diabetic and non-diabetic NOD mice.** (a) Histogram showing the number frequency of TCR sequences on the y axis with counted cells > 0 on the x axis. (b) Density plot showing the relative frequency of single TCR sequences found in greater than 2 cells. Heatmaps showing the relative number of T cells for each TCR by row over the (c) sample and (d) tissue in columns. (e) Heatmap showing the frequency of TCR clones that have more than 10 cells in the entire dataset. Color bars on the left represent the group identity of the TCR clone that is used for differential expression analysis in Supplementary File 14. (f) The number of cells per sample is plotted for all cells belonging to each group of TCR sequences. The color represents which sample type and tissue the number of cells are calculated from. (g) Heatmap showing the relative expression of exhaustion marker genes in group 3 CD4 + and DN T cells. The color bars labeled type show the category of each cell's cluster identity. Exhausted cell clusters are more associated with a higher level of expression of exhaustion markers.
(TIF)

**S9 Fig. Lineage tracing and expression of infiltrating T cell clones.** (a) Per TCR monocle trajectories plotted for each cell in the diabetic (left) and non-diabetic (right) mice. Each line represents the trajectory of one TCR clone across UMAP positions. (b) Volcano plot showing differential expression of genes in the infiltrating vs non-infiltrating T cells in diabetic mice with positive log fold change in the infiltrating T cells. Significant log fold change values > 1 and adjusted P-values < 0.01 are colored red and top 25 significant genes are labeled. (c) Violin and box plot showing average expression of Ifng between tissue matched (infiltrating) and non-matched T cells. (d) Likelihood estimation for bootstrapped pair transitions within clones. Each clone was randomly sampled for pairs of cells 100 times and the likelihood of each transition was calculated based on the proportion of two different cell types being chosen. The results were plotted as a graph with the nodes representing each cell cluster. The thickness of each bar represents the average percent likelihood of a transition occurring across clones with more than 3 cells. Red colored lines represent transitions occurring within the same compartment (i.e., CD4 to CD4) while blue colored lines represent transitions occurring between compartments (i.e., DN to CD4). The direction of transition cannot be inferred however the nodes are organized from bottom to top in order of ascending T cell maturation. (e) Epitope targets were predicted from the VDJdb database and cells with exact matches were plotted on the cell UMAP and colored by target epitope. (f) Normalized cell numbers were plotted based on targeted epitope by tissue.
(TIF)

**S10 Fig. Expression of TCR genes by cluster.** Dot plot showing cell percentage and average expression in each cluster for tcr alpha (a), beta (b), and gamma/delta (c). (d-f) Psdobulk expression plots showing average expression of tcr gene expression by cluster.
(TIF)

**S1 File. S1 Table.** Sample metadata for individual mice. **S2 Table.** CD3+ + T cell recovery and sequence yield from 10X single-cell library. **S3 Table.** Expression of marker genes across T cell clusters. **S4 Table.** Results of find all markers reported with average Log Fold Change, percent expression of cells, Gene name and both identified and simplified cluster name. **S5 Table.** Results of find markers show differential expression of genes in Naïve populations of DN T cells compared to CD4 T cells. **S6 Table.** Results of find markers show differential expression of genes in IL7r negative DN T cells compared to IL7r- CD4 T cells. **S7 Table.** Results of find markers show differential expression of genes in a mixed DN and CD4 effector T cell

population compared to IL7 positive CD4 Teff and effector Tregs. **S8 Table.** Results of find markers show differential expression of genes in a mixed DN and CD4 effector T cell population compared to IL7 positive DN effector T cells. **S9 Table.** Results of find markers show differential expression of genes in IL7r negative DN T cells compared to Il7r+ + DN T cells. **S10 Table.** Results of find markers show differential expression of genes in Gzma positive CD8 T cells compared to Gzma negative CD8 T cells. **S11 Table.** Table showing GO enrichment of differentially expressed genes between populations of mixed DN and CD4 effector T cells and CD4 effector Tcells and Tregs. **S12 Table.** Table showing GO enrichment of differentially expressed genes between populations of Gzma positive CD8 T cells and Gzma negative CD8 T cells. **S13 Table.** Frequency of T cell clones separated by cluster appearance. Clusters refer to simple classification. Rows listed by specific TCRab clone CDR3 sequence. **S14 Table.** Differential expression of genes in clones specific to each clustered group. Fold change and Pct 1 refer to the group in the group column and pct 2 represents the proportion of expressing cells in all clones from other groups. **S15 Table.** Infiltrating (matching) TCR clones counted by sample type. Alpha beta TCR sequence for CDR3 region is reported. **S16 Table.** Differential expression of genes between islets-matching cells in blood and blood-matching cells in islets. The positive fold change shows enriched blood. **S17 Table.** Differential expression of genes between islets-matching and non-matching cells in blood. Positive Log fold-change shows enriched in matching. **S18 Table.** Differential expression of genes between blood-matching and non-matching cells in islets. Positive log fold-change. **S19 Table.** Exact matches between TCR-seq predicted CDR3 regions and experimentally validated TCR-pMHC interactions in the VDJdb.
(XLSX)

## Acknowledgments

We acknowledge Jennifer Hollister-Lock and the Joslin Islet Isolation Core for helping the purification of mouse islets. We are thankful to Qiong Zhou at Joslin Molecular Phenotyping and Genotyping Core and Stephen Wood at 10X Genomics for providing support during single-cell library preparation. We acknowledge Joslin animal physiology core for the support during NOD mice rearing. MZI acknowledges support for postdoctoral fellowships from the Lundbeck Foundation, Copenhagen, Denmark.

## Author contributions

**Conceptualization:** Md Zohorul Islam, Jacob Luber, Aleksandar Kostic, Michael Robben.

**Data curation:** Md Zohorul Islam, Sam Zimmerman, Aleksandar Kostic, Michael Robben.

**Formal analysis:** Md Zohorul Islam, Sam Zimmerman, Alexis Lindahl, Jacob Luber, Aleksandar Kostic, Michael Robben.

**Funding acquisition:** Jacob Luber, Aleksandar Kostic, Michael Robben.

**Investigation:** Md Zohorul Islam, Sam Zimmerman, Alexis Lindahl, Jacob Luber, Aleksandar Kostic, Michael Robben.

**Methodology:** Md Zohorul Islam, Sam Zimmerman, Jacob Luber, Aleksandar Kostic, Michael Robben.

**Project administration:** Md Zohorul Islam, Jose Ordovas-Montanes, Jacob Luber, Aleksandar Kostic, Michael Robben.

**Resources:** Jose Ordovas-Montanes, Jacob Luber, Aleksandar Kostic, Michael Robben.

**Software:** Sam Zimmerman, Jacob Luber, Aleksandar Kostic, Michael Robben.

**Supervision:** Jon Weidanz, Jose Ordovas-Montanes, Jacob Luber, Aleksandar Kostic, Michael Robben.

**Validation:** Md Zohorul Islam, Alexis Lindahl, Michael Robben.

**Visualization:** Md Zohorul Islam, Sam Zimmerman, Alexis Lindahl, Jacob Luber, Michael Robben.

**Writing – original draft:** Md Zohorul Islam, Jacob Luber, Aleksandar Kostic, Michael Robben.

**Writing – review & editing:** Md Zohorul Islam, Jacob Luber, Aleksandar Kostic, Michael Robben.

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
