## [Decision Letter · Decision Letter 0]

30 Oct 2024

PONE-D-24-38697Single-cell RNA-seq reveals disease-specific CD8+ T cell clonal expansion and a high frequency of transcriptionally distinct double-negative T cells in diabetic NOD micePLOS ONE

Dear Dr. Robben,

Thank you for submitting your manuscript to PLOS ONE. After careful consideration, we feel that it has merit but does not fully meet PLOS ONE’s publication criteria as it currently stands. Therefore, we invite you to submit a revised version of the manuscript that addresses the points raised during the review process.

**We find that the manuscript requires increased precision in several areas, with certain data needing further completion and reinforcement. We invite you to submit a revised version that thoroughly addresses all points raised during the review process (refer to comments from the reviewer and academic editor below).**

We look forward to receiving your revised manuscript.

Kind regards,

Prof. Pierre Bobé

Academic Editor

PLOS ONE

**Journal Requirements:**

2. To comply with PLOS ONE submissions requirements, in your Methods section, please provide additional information regarding the experiments involving animals and ensure you have included details on (a) methods of sacrifice, (b) methods of anesthesia and/or analgesia, and (c) efforts to alleviate suffering.

This study was supported by grants from the Beatson Foundation (Grants ID 2800013). The salary of the author MZI was supported by Lundbeck Foundation, Copenhagen, Denmark (Grants ID R288-2018-1123).

5. Please note that your Data Availability Statement is currently missing the direct link to access each database. If your manuscript is accepted for publication, you will be asked to provide these details on a very short timeline. We therefore suggest that you provide this information now, though we will not hold up the peer review process if you are unable.

7. We note that you have included the phrase “data not shown” in your manuscript. Unfortunately, this does not meet our data sharing requirements. PLOS does not permit references to inaccessible data. We require that authors provide all relevant data within the paper, Supporting Information files, or in an acceptable, public repository. Please add a citation to support this phrase or upload the data that corresponds with these findings to a stable repository (such as Figshare or Dryad) and provide and URLs, DOIs, or accession numbers that may be used to access these data. Or, if the data are not a core part of the research being presented in your study, we ask that you remove the phrase that refers to these data.

**Additional Editor Comments:**

1) In the discussion section (line 372), it should be clarified that the *lpr*  mutation in the lupus-prone MRL genetic background (MRL/*lpr* ) or in the normal B6 genetic background (B6/*lpr* ) does not result from a null mutation in the *fasL* gene as indicated in the discussion. Instead, it arises due to the insertion of the retrotransposon ETn in the second intron of the *fas*   gene, leading to transcriptional repression. Therefore, *lpr*  mice, like NOD mice, possess functional copies of *fasL genes* . Additionally, DN T cells and CD8+ T cells from MRL/*lpr*  or B6/*lpr*  mice exhibit elevated expression of functional FasL on the plasma membrane. Furthermore, to the best of my knowledge, the current understanding is that pathogenic DN T cells in *lpr*  mice are mainly generated in the periphery as a result of dysregulated homeostasis of effector/memory T cells.

2) The authors should provide a detailed explanation of their methodology to demonstrate that the DN cells observed in NOD mice arise from their own proliferation, as stated line 403 (conclusion), rather than from the conversion of CD8+ cells into DN cells. Indeed, DN T cells could originate from CD8+ T cells, as shown in various lupus mouse models where chronic activation leads CD8+ T cells to downregulate their CD8 molecules. Therefore, have you quantified CD8 surface levels (MFI) in NOD CD8+ T cells using flow cytometry? Additionally, do DN T cells in NOD mice overexpress the phosphatase B220 (CD45RABC) along with Fas ligand, similar to DN T cells in lupus mouse models?

Reviewers' comments:

Reviewer's Responses to Questions

**Comments to the Author**

1. Is the manuscript technically sound, and do the data support the conclusions?

Reviewer #1: Yes

2. Has the statistical analysis been performed appropriately and rigorously? 

Reviewer #1: Yes

3. Have the authors made all data underlying the findings in their manuscript fully available?

Reviewer #1: Yes

4. Is the manuscript presented in an intelligible fashion and written in standard English?

Reviewer #1: Yes

5. Review Comments to the Author

**Reviewer #1: ** Dears,

I read with great interest and attention the manuscript “Single-cell RNA-seq reveals disease-specific CD8+ T cell clonal expansion and a high frequency of transcriptionally distinct double-negative T cells in diabetic NOD mice.” This study is original and relevant to the field, highlighting, through single-cell RNA sequencing, T cell subsets infiltrating the islets and potentially important T cell receptor sequences linked to the pathogenesis of type 1 diabetes.

Overall, the manuscript is well-constructed and fits within the journal's scope. The findings are interesting, properly interpreted, and well-discussed. Using robust and reproducible immunological methods, the authors demonstrate a high frequency of double-negative T cells throughout type 1 diabetes pathogenesis, as well as CD8+ T cells infiltrating with exhausted, cytotoxic, and inflammatory states during disease development

RECOMMENDATION: Publish after MINOR ADJUSTMENTS:

1) In the Materials and Methods section "Blood glucose measurement and diabetes diagnosis" (line 433), I suggest including a reference for the parameters used to diagnose diabetic mice (two consecutive glucose readings of ≥250 mg/dl considered diabetic).

2) Figure 1D, results: Is it possible to evaluate the frequency of T cells in the islets of non-diabetic mice using flow cytometry?

3) Considering the complementary determining region (CDR3) of autoreactive T cell clones, it could be interesting to determine the antigen fragments potentially recognized using the BLAST tool. This could reinforce that, in particular, Trav16n, Trbv29, and Trav8d-2 exhibit CDR3 regions that may potentially recognize insulin peptides.

6. PLOS authors have the option to publish the peer review history of their article (what does this mean? ). If published, this will include your full peer review and any attached files.

**Do you want your identity to be public for this peer review?** For information about this choice, including consent withdrawal, please see our Privacy Policy .

Reviewer #1: No

---

## [Author Response · Author response to Decision Letter 1]

26 Nov 2024

All concerns and comments from the reviewer and editor have been addressed in the main manuscript and are addressed individually in the "Response to Reviewers" document that has been submitted with the revised manuscript. Please let us know if anything else is required for publication consideration.

---

## [Editor Report · Decision Letter 1]

26 Dec 2024

PONE-D-24-38697R1Single-cell RNA-seq reveals disease-specific CD8+ T cell clonal expansion and a high frequency of transcriptionally distinct double-negative T cells in diabetic NOD mice

PLOS ONE

Dear Dr. Robben,

Thank you for submitting your manuscript to PLOS ONE. After careful consideration, we feel that it has merit but does not fully meet PLOS ONE’s publication criteria as it currently stands. Therefore, we invite you to submit a revised version of the manuscript that addresses the points raised during the review process.

The authors have addressed many of the issues that emerged during the review process.

However, some minor points persist in the text, such as the inconsistent spelling of "pseudo," which is incorrectly written as "psuedo" at least six times.Furthermore, I recommend that the authors include the FSC-A vs. SSC-A dot plot in their gating strategy (Supplementary Figure 5) to demonstrate that only single cells are included in the flow cytometry analysis, while doublets or multiplets are excluded.I found the statements regarding Fas-deficient lupus mice in the Discussion section on page 20 to be ambiguous in the first instance and inaccurate in the second: “Studies in the MRL-lpr mice ….. have connected lymphoproliferation of DN T cells to a reduced apoptotic signal of negatively selected T cells in the thymus [50], as well as “ …….would suggest peripheral development of DN T cells not seen in lpr mice. There is substantial evidence indicating that peripheral DN T cells are generated in the periphery of lpr mice and patients with autoimmune lymphoproliferative syndrome (ALPS), rather than representing primitive αβ T cells that escape late-stage thymocyte development before migrating to peripheral tissues. Specifically, multiple reports have shown that B220^+^ DN T cells originate from CD8^+^  T cells at the effector/memory stage following repeated TCR engagement in Fas-deficient lupus mice. Similar findings have also been reported in human studies.To avoid any confusion for the readers, I recommend adding "SP" when referring to the CD4+ and CD8+ phenotype, for instance, CD4/DN and CD4+/CD8+ could be written CD4+ SP/DN and CD4+/CD8+ SP T cells.Additionally, when mentioning the MRL/lpr mouse strain, I suggest including the term "lupus-prone" (e.g., lupus-prone MRL/lpr mice) for those who may not be familiar with this mouse model of systemic lupus erythematosus (page 9).

We look forward to receiving your revised manuscript.

Kind regards,

Prof. Pierre Bobé

Academic Editor

PLOS ONE

Journal Requirements:

Additional Editor Comments:

The spelling of "pseudo," appears incorrect at least six times as "psuedo."Furthermore, I recommend that the authors include the FSC-A vs. SSC-A dot plot in their gating strategy (Supplementary Figure 5) to demonstrate that only single cells are included in the flow cytometry analysis, while doublets or multiplets are excluded.I found the statements regarding Fas-deficient lupus mice in the Discussion section on page 20 to be ambiguous in the first instance and inaccurate in the second: “Studies in the MRL-lpr mice ….. have connected lymphoproliferation of DN T cells to a reduced apoptotic signal of negatively selected T cells in the thymus [50], as well as “ …….would suggest peripheral development of DN T cells not seen in lpr mice. There is substantial evidence indicating that peripheral DN T cells are generated in the periphery of lpr mice and patients with autoimmune lymphoproliferative syndrome (ALPS), rather than representing primitive αβ T cells that escape late-stage thymocyte development before migrating to peripheral tissues. Specifically, multiple reports have shown that B220^+^ DN T cells originate from CD8^+^  T cells at the effector/memory stage following repeated TCR engagement in Fas-deficient lupus mice. Similar findings have also been reported in human studies.To avoid any confusion for the readers, I recommend adding "SP" when referring to the CD4+ and CD8+ phenotype, for instance, CD4/DN and CD4+/CD8+ could be written CD4+ SP/DN and CD4+/CD8+ SP T cells.Additionally, when mentioning the MRL/lpr mouse strain, I suggest including the term "lupus-prone" (e.g., lupus-prone MRL/lpr mice) for those who may not be familiar with this mouse model of systemic lupus erythematosus (page 9).

---

## [Author Response · Author response to Decision Letter 2]

6 Jan 2025

Additional comments and review have been addressed in the file titled "Corrected_NOD_PlosONE_author_responses_1_6_25".

---

## [Editor Report · Decision Letter 2]

9 Jan 2025

Single-cell RNA-seq reveals disease-specific CD8+ T cell clonal expansion and a high frequency of transcriptionally distinct double-negative T cells in diabetic NOD mice

PONE-D-24-38697R2

Dear Dr. Robben,

We’re pleased to inform you that your manuscript has been judged scientifically suitable for publication and will be formally accepted for publication once it meets all outstanding technical requirements.

Kind regards,

Prof. Pierre Bobé

Academic Editor

PLOS ONE
---

## [Editor Report · Acceptance letter]

PONE-D-24-38697R2

PLOS ONE

Dear Dr. Robben,

I'm pleased to inform you that your manuscript has been deemed suitable for publication in PLOS ONE. Congratulations! Your manuscript is now being handed over to our production team.

Kind regards,

on behalf of

Prof Pierre Bobé

Academic Editor

PLOS ONE
